# HHV8-Negative Effusion-Based Large B Cell Lymphoma Arising in Chronic Myeloid Leukemia Patients under Dasatinib Treatment: A Report of Two Cases

**DOI:** 10.3390/biology10020152

**Published:** 2021-02-14

**Authors:** Stefano Fiori, Elisabetta Todisco, Safaa Ramadan, Federica Gigli, Patrizia Falco, Alessandra Iurlo, Cristiano Rampinelli, Giorgio Croci, Stefano A. Pileri, Corrado Tarella

**Affiliations:** 1Division of Diagnostic Hematopathology, European Institute of Oncology, IRCCS, Via Ripamonti 435, 20141 Milan, Italy; stefano.pileri@ieo.it; 2Division of Onco-Hematology, European Institute of Oncology, IRCCS, Via Ripamonti 435, 20141 Milan, Italy; Elisabetta.Todisco@ieo.it (E.T.); safaa.ramadan@ieo.it (S.R.); Federica.Gigli@ieo.it (F.G.); Corrado.Tarella@ieo.it (C.T.); 3Department of Medical Oncology, NCI-Cairo University, Kasr Al Eini Street, 11796 Cairo, Egypt; 4SSD Ematologia, ASLTO4, Ospedali di Chivasso Cirié Ivrea, TO10034 Chivasso, Italy; PFalco@aslto4.piemonte.it; 5Division of Hematology, Foundation IRCCS Ca’ Granda Ospedale Maggiore Policlinico, 20122 Milan, Italy; alessandra.iurlo@policlinico.mi.it; 6Division of Medical Imaging and Radiation Sciences, IEO, European Institute of Oncology, IRCSS, 20141 Milan, Italy; cristiano.rampinelli@ieo.it; 7Division of Pathology, Foundation IRCCS Ca’ Granda Ospedale Maggiore Policlinico, 20122 Milan, Italy; Giorgio.Croci@unimi.it; 8Dipartimento Universitario di Scienze della Salute (DISS), Universita’ di Milano, 20133 Milan, Italy

**Keywords:** CML, TKIs, large B cell lymphoma, EBL, pleural effusion, dasatinib, EBV, HHV8, large B-cell effusion-based lymphoma

## Abstract

**Simple Summary:**

Tyrosine kinase inhibitors (TKIs) are the treatment of choice for BCR-ABL1-positive chronic myeloid leukemia (CML). Dasatinib is a second generation TKI frequently associated with pleural effusion in up to 33% of patients. Here, we describe two cases of HHV8-negative large B cell effusion-based lymphoma (EBL) confined to the pleura, incidentally, diagnosed in patients presenting with dasatinib-related pleural effusion. One patient is alive and is in remission at 17 months from diagnosis, while unfortunately the other patient died of progressive disease and novel coronavirus disease 2019 (COVID-19)-related pneumonia 16 months from diagnosis. These cases of large B cell EBL in patients receiving dasatinib raise concern about a possible association and we strongly recommend cytological investigation in patients with persistent/relapsing pleural effusion under dasatinib to improve the knowledge about this entity.

**Abstract:**

Tyrosine kinase inhibitors (TKIs) are the treatment of choice for BCR-ABL1-positive chronic myeloid leukemia (CML). Although TKIs have substantially improved prognosis of CML patients, their use is not free of adverse effects. Dasatinib is a second generation TKI frequently associated with pleural effusion in up to 33% of patients. This results in symptoms as dyspnea, cough and chest pain that may require therapy discontinuation. In the present report, we describe two exceptional cases of HHV8-negative large B-cell effusion-based lymphoma (EBL) confined to the pleura, incidentally, diagnosed in patients presenting with dasatinib-related pleural effusion. One patient (case 1) is alive and is in remission at 17 months from large B-cell EBL diagnosis while unfortunately the other patient (case 2) died of progressive disease and COVID-19 pneumonia 16 months from large B-cell EBL diagnosis. These cases raise concern about a possible association between large B-cell EBL and dasatinib, and the different clinical outcome of the two cases poses a challenge in treatment decision. For this reason, we strongly recommend cytological investigation in patients with persistent/relapsing pleural effusion under dasatinib, primarily to validate its possible association with lymphoma development and to improve the knowledge about this entity.

Tyrosine kinase inhibitors (TKIs) are the treatment of choice for *BCR-ABL1*-positive chronic myeloid leukemia (CML). Although TKIs have substantially improved prognoses of CML patients, their use is not free of adverse effects.

Dasatinib is a second generation TKI frequently associated with pleural effusion in up to 33% of patients. This results in symptoms such as dyspnea, cough, and chest pain, that may require discontinuation of therapy. Cardiac disease, hypertension, advanced age, and duration of treatment are the most common risk factors for dasatinib-related pleural effusion [1]. Pleural effusion is also associated with bosutinib, another second generation TKI, particularly in patients who developed pleural effusion during previous dasatinib treatment [2].

Lymphoproliferative disorders (LPDs) are exceptional side effects during CML treatment with TKIs [3,4,5,6,7]. Although clonal expansion of T-large granular lymphocytes (T-LGL) [8] and reactive follicular hyperplasia [9] were reported with dasatinib, overt LPDs are extremely rare and have particularly been described in association with first generation TKI imatinib [3,4,5,6,7].

Here, we describe two cases of HHV8-negative large B cell effusion-based lymphoma (EBL) confined to the pleura, incidentally diagnosed in patients presenting with dasatinib-related pleural effusion. One patient (case 1) is alive and is in remission at 17 months from large B cell EBL diagnosis (case 1), while unfortunately the other patient (case 2) died of progressive disease and novel coronavirus disease 2019 (COVID-19)-related pneumonia 16 months from large B cell EBL diagnosis.

Case 1: a 69-year-old male, with history of hypertension, was diagnosed with CML in 1998, with a Sokal and Hasford score as relative low risk (RR) and was continuously treated with imatinib from 1999 to 2011, obtaining deep molecular remission (DMR) after six months. For a CML morphologic relapse, he was switched to dasatinib in 2011 and DMR was achieved again. Since 2012, he had experienced mild recurrent pleural effusions, managed with steroids and dasatinib dose modulation. The patient had never been symptomatic and CML was in persistent DMR; therefore, dasatinib treatment was continued. In June 2019, loss of DMR was detected (>0.1% IS), with persistence of pleural effusion. The patient was then considered for the phase II prospective randomized trial comparing ABL001 versus bosutinib (ClinicalTrials.gov identifier: NCT03106779). During the screening phase, the pleural effusion was drained, and cytologic examination showed hematic effusion containing large atypical lymphoid cells, with immunoblastic morphology and B cell phenotype (positive for CD20, PAX5, CD97a), positive for Epstein–Barr virus (EBV) in situ hybridization (EBER), and with high Ki67 proliferation index. The cells were positive for IRF4, partially for BCL6, CD30, BCL2 and C-MYC and negative for HHV8, CD10. A molecular study for B cell clonality showed monoclonal rearrangement of immunoglobulin heavy chain genes (*IGH*), confirming the clonal nature (Figure 1). FISH analysis for BCR-ABL1 was negative. Peripheral cell blood count was normal, with the exception of mild lymphocytopenia (700/mmc). Subsequent total-body computed tomography (CT) showed bilateral pleural effusion with thickening of the right parietal pleura (Figure 2), and ^18^FDG positron emission tomography (PET) highlighted only a weak uptake (SUV 3) in the right pleura. Bone marrow evaluation was negative for a lymphoproliferative disease, and confirmed CML morphologic and cytogenetic remission, while *BCR-ABL1* transcript was >0.1% IS. Peripheral blood polymerase chain reaction (PCR) for EBV-DNA revealed 1300 copies/mmc. IgM and IgG anti- Viral capsid antigen (VCA)titers were consistent with previous infection. A diagnosis of EBV-positive-large B cell effusion-based lymphoma (EBL) was made.

In consequence of the new diagnosis and a screening failure, bosutinib was started at full dose. Despite the aggressive histologic features of the disease, the patient was treated with six weekly rituximab administrations, along with strict thoracic ultrasonography monitoring of the pleural thickening. With this strategy, the patient achieved a very good local response, in absence of any extrapleural disease localization.

After two months of bosutinib, the patient developed severe liver toxicity (G3) that imposed dose reduction. After six months of half-dose bosutinib, the patient had severe pleural effusion recurrence which resolved after bosutinib withdrawal, diuretic, and steroid treatment. Pleural effusion cytologic evaluation was performed and showed no evidence for lymphoma. Immediately after, CML morphological relapse occurred. Mutational analysis showed an absence of compound mutations, and ponatinib was started, obtaining morphologic and cytogenetic remission after three months then DMR at six months. The 16-month follow up CT scan shows a stable minimal thickening of the right pleura, without pleural effusion and any related symptoms, consistent with persistent complete remission (CR) (Figure 2).

Case 2: a 69-year-old male patient, diagnosed with CML in 2015, with an intermediate Sokal RR score and low Hasford RR score. The patient was a former smoker and had a history of cerebellar ischemia in 2003. The patient received first-line dasatinib, obtaining persistent DMR after three months of treatment. In August 2019, while under dasatinib, he experienced chest pain, and a chest CT scan showed a right subpleural lung nodule (2.3 cm), with thickening of the homolateral major pleural fissure and left pleural effusion (Figure 2). ^18^FDG-PET showed diffuse metabolic activity in the right pleura, with an accentuated area (SUV 4.9) corresponding to the lung nodule in the lower right lobe. A right pleural decortication and pulmonary segmentectomy of the right lower lobe in thoracoscopy were performed, and a left pleural drainage tube was inserted.

Histologic evaluation of the lung nodule showed squamous cell carcinoma. Cytologic examination of pleural fluid showed sheets of large atypical lymphoid cells with B cell phenotype (positive for CD20, PAX5, CD79a), partially positive for IRF4, BCL6, CD30, BCL2 and C-MYC, negative for EBV/EBER, HHV8, CD10, and for the plasma cell markers CD138 and EMA and high Ki67 proliferation index. The same lymphoid cells were focally present in the pleural biopsy. Molecular study for B cell clonality showed monoclonal rearrangement of *IGH* (Figure 1). FISH analysis for BCR-ABL1 was negative, and a diagnosis of large B cell EBL was concluded. Peripheral blood cell counts showed only mild lymphopenia (600/mmc), and PCR analysis for EBV was negative. Subsequently, a total-body ^18^FDG-PET/CT scan and bone marrow evaluation were negative for lymphoma, with persistent DMR of the CML. Although the patient fulfilled the criteria for dasatinib discontinuation, we decided to shift to imatinib. This decision was conditioned by the current COVID-19 pandemic, with limited access to regular hospital and laboratory services, in particular to frequent BCR-ABL molecular determination.

Considering the absence of any sign of lymphoma at the post-intervention PET/CT scan, in a patient with multiple tumors, we opted for a strict clinical and radiologic follow up. The CT scan evaluation after three months did not demonstrate any signs of disease progression.

Meanwhile, the COVID-19 pandemic was escalating in Italy, and unfortunately a close follow up was not feasible. In addition, the patient, who was totally asymptomatic, chose to postpone the radiologic evaluation.

Approximately six months from the last visit, the patient presented again with a persistent irritating cough and shortness of breath. A PET/CT scan showed multiple lymphoadenopathies at the mediastinal (13 × 5 cm ²), cephalo-pancreatic, lumboaortic abdominal levels (max 3 cm) and pericardial effusion. Meanwhile, the patient remained in DMR of the CML. The patient was urgently admitted, and a mediastinal lymph-node biopsy was performed by endobronchial ultrasound (EBUS). Histological evaluation confirmed diffuse large B cell lymphoma, for which debulking chemotherapy with a CHOP-like regimen was started. Immediately, the patient developed respiratory insufficiency and was transferred to the intensive care unit, and COVID-19 pneumonia was diagnosed. Unfortunately, after 26 days in an intensive care unit, the patient died due to respiratory failure.

LPDs are rarely described during TKIs treatment in CML patients, specifically with imatinib [3,4,5,6,7]. EBV-positive LPDs are reported in only three CML imatinib-treated patients [6,7,8]. The first described EBV-positive LPD patient had an isolated skin lesion, with LBCL-like histology, that regressed after imatinib dose reduction [6]. The second patient, also with LBCL-like lymphoma, had nodal, bone marrow, splenic and hepatic involvement. In this case, the lymphoma did not regress with imatinib discontinuation and rituximab treatment, but required the addition of chemotherapy (CHOP) to obtain CR [7]. The third case had nodal and splenic involvement with polymorphous histologic features. In this case, CR was achieved with non-intensive therapy (methylprednisolone, cyclophosphamide, and rituximab), given the patient’s frail status [8].

The development of EBV-positive LPDs in subjects under TKI treatment can be explained by the associated T cell suppressive effect. In vitro studies have reported that TKIs interfere with T cell receptor activation by phosphorylating downstream kinases as ZAP70 and LAT and reducing the antigen-triggered expansion of CD8-positive cells, that are crucial in controlling viral infection, including EBV reactivation [10]. A specific effect of dasatinib is the “blockage” of CCR7-positive terminal effector T lymphocytes in peripheral blood, by impairment of their migration in tissues [11]. Histologic and clinical features of the three described EBV-positive LPD cases occurring under imatinib strengthen the T cell suppressive effect of TKIs.

In fact, all three EBV-positive LPD patients described in the literature had peripheral T cell depletion, with a reduction in cytotoxic T lymphocytes. In the first case [6], the isolated skin lesion suggested an EBV-positive mucocutaneous ulcer, newly recognized entity associated with immunosuppression and immunosenescence [12]. The polymorphic histologic features of the third case of reminiscent of polymorphic post-transplant LPD were typically linked to post-transplant immunosuppression [13].

By contrast, the few reported EBV/HHV8-negative LBCLs arising during imatinib did not resolve with TKI discontinuation, but required active treatment [3,4].

Kaji, D. et al. reported the results of a retrospective analysis of a series of HHV8-negative primary effusion-based lymphoma, demonstrating a favorable outcome with anthracycline based chemotherapy. This report, however, did not provide any information on the concurrent CML treatment [14].

In the present report, we describe two exceptional cases of large B cell EBL arising under dasatinib treatment, as the first report of two consecutive cases in a single center after the first described patient in 2017, and the second single case just published late 2020 [15,16]. In the two patients, lymphoma was limited to the pleura at the time of diagnosis, therefore other subtypes of LPDs primarily presenting as pleural effusion had to be excluded. Negativity for HHV8 ruled out primary effusion lymphoma. Moreover, both described cases had strong expression of B cell antigens (CD20, PAX5, CD79a), and were negative for the plasma cell markers CD138 and EMA. This phenotype is different from the one characteristically found in HHV8-positive primary effusion lymphoma, which shows expression of plasma cell markers along with the down-regulation or loss of B cell markers. None of the two described patients had a fluid overload condition, in contrast with a previous report by Alexanian et al. [17]. In case 1, positivity for EBV and incidental findings might suggest fibrin-associated LBCL [18], but the lack of fibrin in the pleural effusion (fluid and hematic) did not support such a diagnosis.

In agreement with the reported T cell suppressive effect of TKIs, both patients had lymphocytopenia at the time of LPD diagnosis.

In conclusion, these two cases of large B cell EBL in patients receiving dasatinib raise concern about a possible association. The different clinical outcome of the two patients poses a challenge in treatment decision. This is in part due to the lack of a defined histologic entity and the uncertainty of an association with dasatinib treatment. For this reason, we strongly recommend cytological investigation in patients with persistent/relapsing pleural effusion under dasatinib, primarily to validate its possible association with lymphoma development and to improve the knowledge about this entity.

## Figures and Tables

**Figure 1 biology-10-00152-f001:**
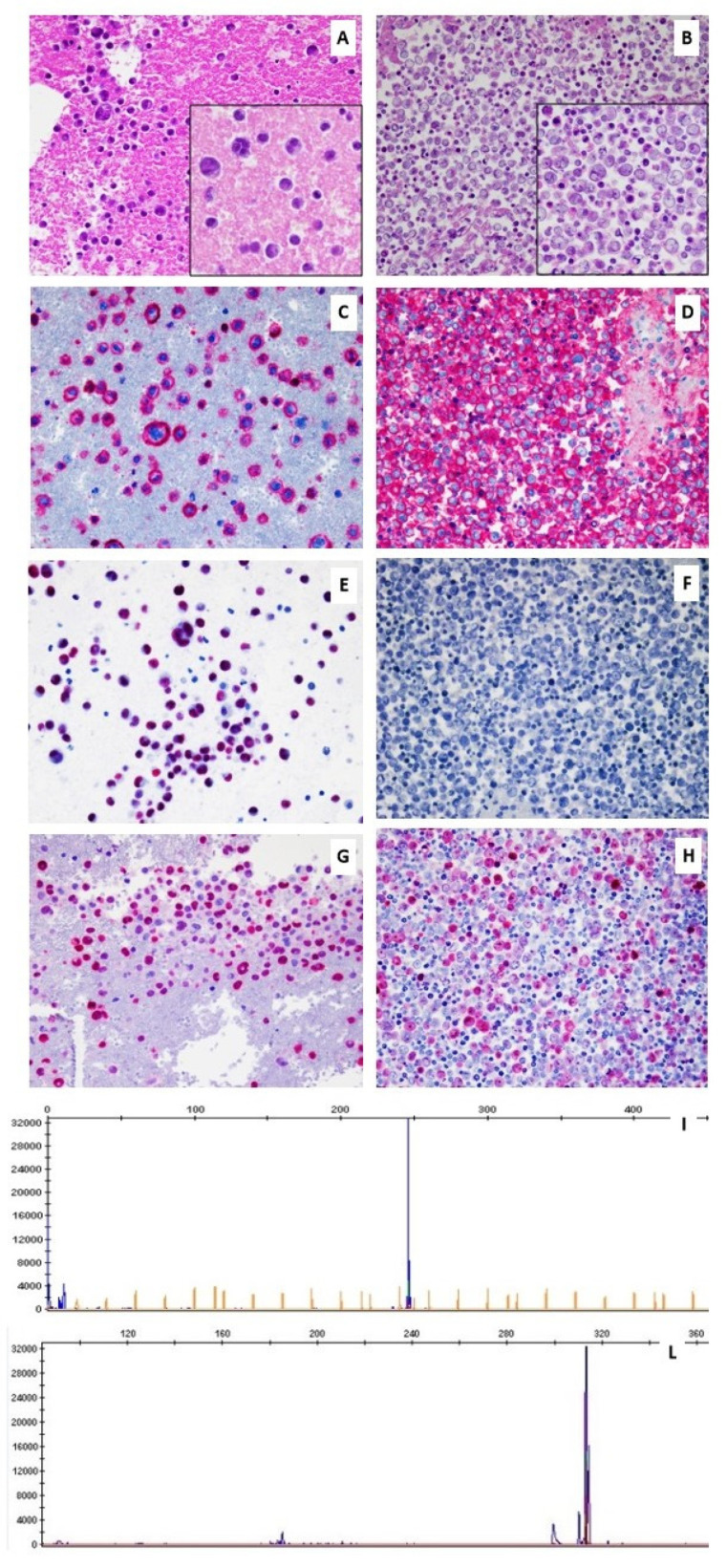
Histopathological features and *IGH* rearrangement of the two cases of LBCL-like effusion-based lymphoma (EBL) diagnosed in dasatinib-related pleural effusion. Case 1 shows the hematic background including immunoblastic cells (**A**), positive for CD20 (**C**), EBV/EBER (**E**), and with high Ki67 proliferation index (**G**). *IGH* rearrangement is clonal, as indicated by a monoclonal peak in *FR3* region (**I**). Case 2 shows sheets of large lymphoid cells (**B**), positive for CD20 (**D**), negative for EBV/EBER (**F**), and with high Ki67 proliferation index (**H**). Additionally, in this case, a monoclonal peak in the *FR3* region proves the clonal rearrangement of *IGH* (**L**).

**Figure 2 biology-10-00152-f002:**
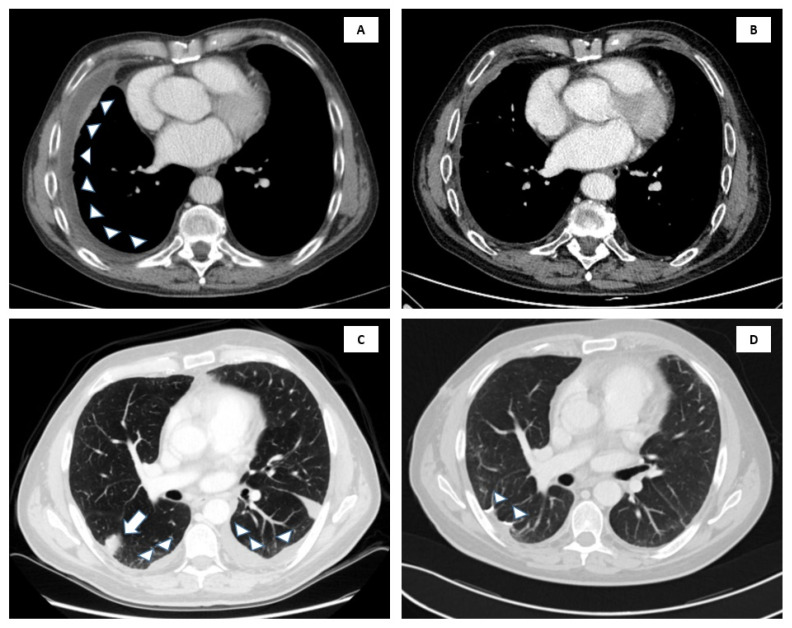
CT scans of the two cases at onset of large B cell effusion-based lymphoma (EBL) and at the last evaluation. In case 1 at onset (**A**), but after drainage procedure of pleural effusion, the right pleura is thickened with minimal pleural effusion (arrowheads). At the last evaluation (**B**), the right pleural thickening and pleural effusion are markedly reduced. In case 2, onset scans (**C**) show bilateral pleural effusion (arrowheads), at the right upper lobe nodule (arrow). At evaluation post-intervention (**D**), note the subpleural suture in the site of wedge resection (arrowheads), and the absence of pleural effusion.

## Data Availability

No new data were created or analyzed in this report, we presented retrospective data for results from routine clinical practice. Data sharing is not applicable to this article.

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
