# Peer review of "HHV8-Negative Effusion-Based Large B Cell Lymphoma Arising in Chronic Myeloid Leukemia Patients under Dasatinib Treatment: A Report of Two Cases"

_biology, 2021, doi:10.3390/biology10020152_

Round 1

Reviewer 1 Report

General evaluation:

The authors describe the course of two patients with chronic myeloid leukemia (CML) who developed small pleural effusion during the treatment with dasatinib. Cytological examination of the fluid incidentally detected neoplastic large B-cells, leading to the diagnosis of effusion-based lymphoma (EBL). One patient favorably responded to single-agent rituximab, whereas the other patient showed progression of lymphoma under the COVID-19 pandemic. The authors suggest cytological examination of pleural fluid for dasatinib-treated CML patients, when they develop persistent/relapsing, even small, pleural effusion. The paper is well written and is of value to the readers in this field.

Comments:

1) It is of interest that both patients showed mild but significant lymphocytopenia. Although the authors attribute this condition to T-cell suppressive effect of TKIs, dasatinib-treated CML patients often experience lymphocytosis, which is associated with higher rates of pleural effusion and improved response rate (Leukemia, 2009;23:1398). This reviewer therefore speculates that the current two patients had underlying immunosuppressive conditions that caused lymphocytopenia, thereby accounting for the development of EBL.

2) Neoplastic cells of HHV8-positive primary effusion lymphoma (PEL) usually lack pan-B-cell markers (i.e., CD19, CD20, and CD79a) and immunoglobulin expression, but variably express activation- and plasma cell-related antigens. Thus, the authors might describe the difference in immunophenotypic features between HHV8-positve PEL and current two cases.

3) Figure 1, L: Monoclonal peak cannot be recognized in this figure.

4) Additional case of dasatinib-related effusion lymphoma has been published (Cytopathology, 2020;31:602-606).

Author Response

We thank you very much for this opportunity and for your appreciation of our work. Your comments enabled us to further improve the quality of our manuscript. In accordance with these suggestions, we modified the manuscript and followed all the requested changes. In the following pages are the responses to the reviewer.

We uploaded a copy of our revised manuscript in which revisions are highlighted by using the track changes mode in MS Word.

Responses:

Reply to comment 1:

We agree with the reviewer that more than 30% of dasatinib treated patients experience lymphocytosis, therefore, we agree that we can’t exclude that the lymphopenia in these two cases might be due to another Dasatinib independent mechanism. However, we would like to clarify that in the discussion we intended to focus on the TKI associated impaired T cell activation and its T-cell suppressive effect as reported in EBV-positive LPDs in subjects under TKI treatment.

Reply to comment 2:

We have added in the manuscript more details about the immunophenotypic features of both cases as can be seen by track changes in page2 for case 1, page 5 for case 2 and in the discussion in page 6.

Reply to comment 3: we have included a new figure that should be more clear, so hopefully this will respond the reviewer comment.

Reply to comment 4: We have added this publication in references (ref 16) together with a comment in the discussion in page 6 and modified the paragraph accordingly as follows: (In the present report, we describe two exceptional cases of large B-cell EBL arising under dasatinib treatment, as the first report of two consecutive cases in a single center after the first described case in 2017 and the second single cases just recently published15,16. )

Reviewer 2 Report

Stefano Fiori and colleagues describe two HHV8-negative large B-cell effusion-based lymphoma (EBL) cases, confined to the pleura, incidentally diagnosed in patients with CML treated with Tyrosine kinase inhibitors (TKIs). The TKIs considered the choice-treatment for CML, presents several side effects, of which pleural effusion is the most common. Although the clonal expansion of T-large granular lymphocytes (T-LGL) and reactive follicular hyperplasia are well-described side-effects of TKIs treatment, the occurrence of lymphoproliferative disorders is exceptional.

This report is attractive and original; the presented cases are the first two HHV8-negative large B-cell effusion-based lymphoma (EBL) associated with TKIs treatment in CML, described in the literature.

This report is well written, easy to read and complete in all its parts: the authors provided a detailed clinical history, representative imaging, well-described morphology and immunophenotype, well supported by the images provided, molecular data, and the complete treatment plan with the outcome.

The references are complete and updated.

Finally, I believe that the topic may be particularly interesting for clinicians who usually deal with this disease and its first-choice therapy: The knowledge of this event, albeit exceptional, allows them a more careful diagnostic evaluation of patients treated with TKIs with persistent or relapsing pleural effusion, not only through the use of imaging but also of cytological investigation for better management of these patients.

Author Response

Reply to reviewer 2:

We are grateful to Reviewer for the time spent in revising our work. We definitively appreciate his/her estimation of our report. We totally agree that this kind of reports raise awareness and attention for careful evaluation of similar cases that might be miss diagnosed, in fact in our conclusion, in agreement to the reviewer, we strongly recommend cytological investigation in patients with persistent/relapsing pleural effusion under dasatinib primarily to validate its possible association with lymphoma development and to improve the knowledge about this entity
